# Current Applications of Artificial Intelligence to Classify Cervical Lymph Nodes in Patients with Head and Neck Squamous Cell Carcinoma—A Systematic Review

**DOI:** 10.3390/cancers14215397

**Published:** 2022-11-02

**Authors:** Matthias Santer, Marcel Kloppenburg, Timo Maria Gottfried, Annette Runge, Joachim Schmutzhard, Samuel Moritz Vorbach, Julian Mangesius, David Riedl, Stephanie Mangesius, Gerlig Widmann, Herbert Riechelmann, Daniel Dejaco, Wolfgang Freysinger

**Affiliations:** 1Department of Otorhinolaryngology-Head and Neck Surgery, Medical University of Innsbruck, 6020 Innsbruck, Austria; 2Department of Radiation-Oncology, Medical University of Innsbruck, 6020 Innsbruck, Austria; 3University Hospital of Psychiatry II, Medical University of Innsbruck, 6020 Innsbruck, Austria; 4Ludwig-Boltzmann Institute for Rehabilitation Research, 1100 Vienna, Austria; 5Department of Radiology, Medical University of Innsbruck, 6020 Innsbruck, Austria

**Keywords:** head and neck neoplasms, head and neck cancer, head and neck squamous cell carcinoma, artificial intelligence, artificial neural networks, machine learning, computed tomography scan, magnetic resonance imaging, positron emission tomography, lymph nodes, lymph node metastases

## Abstract

**Simple Summary:**

Locally-advanced head and neck squamous cell carcinoma (HNSCC) is mainly defined by the presence of pathologic cervical lymph nodes (LNs). Radiologic criteria to classify LNs as pathologic or non-pathologic are shape-based. However, significantly more quantitative information is contained within images. This information could be exploited to classify LNs in patients with locally-advanced HNSCC by means of artificial intelligence (AI). The present work systematically reviews original articles that specifically explore the role of AI to classify LNs in locally-advanced HNSCC. Between 2001 and 2022, 13 retrospective studies were identified. AI’s mean diagnostic accuracy for LN-classification was 86% (range: 43–99%). Consequently, all of the identified studies concluded AI to be a potentially promising diagnostic support tool for LN-classification in HNSCC. However, adequately powered, prospective, randomized control trials are urgently required to further assess AI’s role in LN-classification in locally-advanced HNSCC.

**Abstract:**

Locally-advanced head and neck squamous cell carcinoma (HNSCC) is mainly defined by the presence of pathologic cervical lymph nodes (LNs) with or without extracapsular spread (ECS). Current radiologic criteria to classify LNs as non-pathologic, pathologic, or pathologic with ECS are primarily shape-based. However, significantly more quantitative information is contained within imaging modalities. This quantitative information could be exploited for classification of LNs in patients with locally-advanced HNSCC by means of artificial intelligence (AI). Currently, various reviews exploring the role of AI in HNSCC are available. However, reviews specifically addressing the current role of AI to classify LN in HNSCC-patients are sparse. The present work systematically reviews original articles that specifically explore the role of AI to classify LNs in locally-advanced HNSCC applying Preferred Reporting Items for Systematic Review and Meta-Analysis (PRISMA) guidelines and the Study Quality Assessment Tool of National Institute of Health (NIH). Between 2001 and 2022, out of 69 studies a total of 13 retrospective, mainly monocentric, studies were identified. The majority of the studies included patients with oropharyngeal and oral cavity (9 and 7 of 13 studies, respectively) HNSCC. Histopathologic findings were defined as reference in 9 of 13 studies. Machine learning was applied in 13 studies, 9 of them applying deep learning. The mean number of included patients was 75 (SD ± 72; range 10–258) and of LNs was 340 (SD ± 268; range 21–791). The mean diagnostic accuracy for the training sets was 86% (SD ± 14%; range: 43–99%) and for testing sets 86% (SD ± 5%; range 76–92%). Consequently, all of the identified studies concluded AI to be a potentially promising diagnostic support tool for LN-classification in HNSCC. However, adequately powered, prospective, and randomized control trials are urgently required to further assess AI’s role in LN-classification in locally-advanced HNSCC.

## 1. Introduction

Locally-advanced head and neck squamous cell carcinoma (HNSCC) is defined by advanced tumor and/or nodal stages [1]. The latter is defined by the presence of pathologic cervical lymph nodes (LNs) [1]. Locally-advanced HNSCC patients require multimodality treatment [2], frequently consisting of surgery followed by radiotherapy or primary concurrent radiochemotherapy [3,4]. The presence and the localization of pathologic cervical LNs with or without extracapsular spread (ECS) remain important negative prognostic factors in such patients in terms of overall survival [5,6,7,8].

Besides clinical examination [9,10], the detection of pathologic cervical LNs is currently based on contrast-enhanced computed tomography scans (CTs), magnetic resonance imaging (MRIs) or fluorine-18-fluorodeoxyglucose positron emission tomography scans ([^18^F]FDG-PET) [11,12,13]. Once potentially pathologic cervical LNs are detected [11,12,13], current criteria to classify LNs as pathologic or pathologic with ECS are mainly shape-based [14,15,16,17,18,19]. Mainly, and without further quantification, head and neck radiologists consider LN maximum diameter and margins. Consequently, prognostic implications previously observed for pathologic cervical LNs [5,6,7,8] are based on these shaped-based detection and classifications criteria [14,15,16,17,18,19]. However, significantly more quantitative information about shape, texture, and intensity is contained within CTs, MRIs, and [^18^F]FDG-PET that could be exploited for detection, segmentation, classification, and exploration of prognostic implications of LNs in patients with locally-advanced HNSCC [5], thus, ultimately aiding in clinical decision-making [5].

Radiomic analysis [20,21,22] is an emerging, data-driven technique aiming at extracting, processing, and analyzing such quantitative information from images. The core idea is to treat imaging data, which is frequently obtained during routine clinical staging, as data-mineable sources of information to aid in these processes [5]. This quantitative information contained within images represents a wealth of data for analysis using artificial intelligence (AI). AI, defined as the branch of computer science dedicated to the development of computer algorithms to accomplish tasks traditionally associated with human intelligence [23], may be divided into various subsets. A subset of AI, machine learning (ML) allows computes to learn, through examples, without being programmed with explicit rules [24,25]. Artificial neural networks (ANN) are a subset of ML, which operate on similar functional principles to the human brain neural network, with one input layer of neurons, one or more hidden processing layers of neurons, and one output layer of neurons [26]. Finally, deep learning (DL) is a subset of ANN, which utilizes more than one hidden processing layer of neurons, per definition [26].

The typical radiomic analysis workflow in the context of cervical LNs in locally-advanced HNSCC patients starts with detection, followed by segmentation and classification based on the extracted shape, intensity, and texture features, before prognostic implications are explored [5]. Thus, AI and its respective subsets ML, ANN, and DL may be applied to all of these tasks, ultimately aiming at improving the efficacy of the workflow [5,26].

Currently, various reviews exploring the role of AI in head and neck imaging are available [24,26,27,28]. However, reviews specifically addressing the current role of AI to classify LN in patients with head and neck squamous cell carcinoma are sparse. The present work aims at systematically reviewing original articles applying Preferred Reporting Items for Systematic Review and Meta-Analysis (PRISMA) guidelines [29] and the Study Quality Assessment tool of National Institute of Health (NIH) [30], which specifically explore the current role of AI to classify LNs in patients with locally-advanced HNSCC.

## 2. Materials and Methods

### 2.1. Search Protocol

In the present study, all studies were systematically retrieved that applied (a) AI, ML, ANN, or DL techniques to classify cervical LNs in patients with locally-advanced HNSCC. The systematic search included studies from the databases PubMed [31], Google Scholar [32], and Web of Science [33] between January 2001 and May 2022. The search approach was developed by combining the search keywords: (“Head and Neck Neoplasms” [mesh] OR “head and neck cancer” OR HNC OR HNSCC OR “head and neck squamous cell carcinoma”) AND (“Artificial Intelligence” [mesh] OR “artificial neural networks” OR “machine learning” OR “artificial intelligence”) AND (“computed tomography scan” OR “CT” OR “computed tomography” OR MRI OR “magnetic resonance imaging” OR “positron emission tomography” OR PET) AND (“lymph nodes” [mesh] OR “lymph node“ OR “nodal“ OR “node“ OR lymph node metastasis OR lymph node metastases OR metastasis OR metastases). The potentially relevant articles were exported to EndNote reference manager software (Version 20.2.1; Clarivate Analytics, London, UK) and duplicates were removed. To minimize the possibility of omission of any relevant study, the reference lists of all the eligible articles were manually searched to ensure that all relevant studies were included. Furthermore, the PRISMA guidelines [29] were followed in the searching and screening process (Figure 1). In addition, the corresponding PRISMA checklist [29] was used to ensure that essential aspects of systematic review were considered (Appendix A). This systematic review was not registered.

### 2.2. Inclusion and Exclusion Criteria

The eligible studies were required to have evaluated (a) AI, ML, ANN, or DL techniques to (b) classify LNs in patients with locally-advanced HNSCC. Reviews, case reports, case series, abstracts, studies on animals, conference papers, white papers, editorials, letters to the editor, comments, and expert views were excluded. Studies exploring DNA and RNA microarray genes, proteomics, fluorescence spectroscopy, ultrasound and genetic programming were excluded. In addition, studies in languages other than English and studies exploring thyroid or esophageal cancer were excluded. The details of the inclusion and exclusion process are described in Figure 1.

### 2.3. Screening

To ensure that all eligible studies were included in the present study, a data extraction sheet was used (Microsoft Excel, Redmond, Washington, DC, USA). The data extraction process was conducted by two independent investigators (D.D. and S.M.). Possible discrepancies were resolved by discussion. A consensus was reached on which studies should be included or excluded after deliberations considering the objectives and the inclusion and exclusion criteria of the study.

### 2.4. Parameters Extracted from the Included Studies

The extracted information from each study included author names, year of publication, country, type of study, tumor site, number of HNSCC patients, total number of included cervical LNs and number of cervical LNs considered pathologic, either based on radiologic criteria (“cN+”* or based on histopathologic criteria (“pN+”), number of LNs attributed to training, validation and testing data set, imaging modalities, AI algorithms examined in the study for training data set and test data set with corresponding diagnostic accuracy and conclusion. In addition, the quality of each included study was rated as recommended by the NIH using a binary, 14-item catalogue as “good” (≥11 items rated as “yes”), “fair” (≥8 rated as “yes”) or “poor” (≤7 items rated as “yes”) [30]. Details about the criteria applied to obtain the NIH-score are detailed in Table 1. Additional diagnostic parameters including sensitivity, specificity, and AUC were recorded, if provided, in the data extraction sheet (Table 2).

## 3. Results

### 3.1. Overview of Studies Exploring the Role of Artificial Intelligence in the Classification of Cervcial Lymph Nodes in Patients with Head and Neck Squamous Cell Carcinoma

Between 2001 and 2022, a total of 158 records were identified from the 3 databases PubMed [31], Google Scholar [32], and Web of Science [33]. Of these, 89 records were identified as duplicates and removed. Of the remaining 69 records, another 20 records were manually removed after title-based and abstract-based screening. These excluded records were case reports (*n* = 2), editorials (*n* = 1), letters to the editor (*n* = 1), without available English abstract (*n* = 2), ultrasound as imaging modality (*n* = 2), or malignancies other than HNSCC (*n* = 12). The remaining 49 records were assessed for eligibility based on full text analysis. Based on the full test analysis, another 37 records were removed. Excluded studies were either reviews (*n* = 1), did not apply AI (*n* = 3), did not utilize any imaging modality (*n* = 2), did not explore cervical LNs (*n* = 9), or did not explore cervical LN-classification (*n* = 22).

Notably, the single excluded review was cross-checked for additional literature previously missed, which led to the retrieval of another 22 reviews. These 22 reviews were again cross-checked, ultimately leading to the retrieval of an additional 22 potentially eligible previously unidentified studies. However, all 22 studies were ultimately excluded from the analysis, since these studies did not explore cervical LN-classification applying AI. Finally, one additional study was added based on cross- checking of the 12 originally included studies, which was not retrieved based on the initially defined search terms (Figure 1).

Ultimately, a total of 13 studies that applied artificial intelligence to classify cervical LNs in patients suffering from HNSCC were identified and included. All of the included studies were retrospectively designed and only one study was a multicenter study. The majority of the studies explored patients with oropharyngeal (9 of 13 studies) and oral HNSCC (7 of 13 studies) whereas laryngeal, nasopharyngeal, salivary gland, and unknown primary HNSCC were included rarely (3, 1, 1, and 1 studies, respectively). Of the 13 included studies, 8 explored CTs and 6 PET-CTs. LN segmentation was performed manually in all 13 studies, only 1 study additionally applied additional automatic segmentation. 11 of 13 studies explored the diagnostic accuracy to correctly differentiate pathologic from non-pathologic cervical LNs, and 4 studies additionally explored the capacity to differentiate pathologic cervical LNs with ECS. A total of 9 studies defined histopathologic findings of neck dissection as reference for LN classification. In the remaining 4 studies, LNs were classified based on established radiologic criteria by experienced radiologists, radiation oncologists, or nuclear medicine radiologists. A total of 9 studies applied DL algorithms, the remaining 4 studies applied ML algorithms. The mean number of included patients was 75 patients (standard deviation (SD) ± 72 patients) ranging from 10 to 258 patients. The mean number of included cervical LNs was 340 LNs (SD ± 268) ranging from 21 to 791 LNs. Of these, the mean number of pathologic cervical LNs was 113 (SD ± 76), ranging from 31 to 273 LNs. The mean diagnostic accuracy for training sets was 86% (SD ± 14%) ranging from 43% to 99%, the mean diagnostic accuracy for testing sets was 86% (SD ± 5%) ranging from 76% to 92%. The mean study quality as rated by NIH was 11 of 14 (SD ± 2), ranging from 8 to 13. A total of 9 studies were thus rated as of “good” quality, while 2 studies were rated as “fair” and 2 studies as “poor.” For additional details, please refer to Table 3.

### 3.2. Detailed Presentation of Studies Exploring the Role of Artificial Intelligence in the Classification of Lymph Nodes in Patients with Head and Neck Squamous Cell Carcinoma in Anti-Chronological Order

#### 3.2.1. Benchmarking Eliminative Radiomic Feature Selection for Head and Neck Lymph Node Classification

In 2022, in a monocentric, retrospective study performed in Austria, Bardosi and co-workers explored the diagnostic accuracy of various radiomic eliminative feature selection algorithms for their potential to identify an as small as feasible set of features and yet to retain as much clinical “accuracy” as possible for classifying cervical LNs as non-pathologic, pathologic, or pathologic with ECS [5].

Patients suffering from HNSCC of the oral cavity, pharynx, larynx, and patients suffering from carcinoma of unknown primary treated at their institution with available pretreatment CTs were eligible. All LNs were manually contoured slice-by-slice in axial planes by two experienced head and neck radiologists and labeled as “non-pathologic,” “pathologic”, or “pathologic with ECS” based on established radiologic criteria. Numerous supervised and unsupervised wrapper-type feature selection algorithms were explored. The primary aim of the study was to identify as-small-as-feasible sets of features and yet to retain as much clinical “accuracy” as possible to correctly classify cervical LNs as non-pathologic, pathologic, or pathologic with ECS [5].

At total of 28 patients with 252 LNs were included in the study. Out of these 182 LNs were labeled as “pathologic” and 52 as “pathologic with ECS”. The combination of sparse discriminant analysis and genetic optimization retained up to 90% of the classification accuracy with only 10% of the original numbers of features [5].

The authors concluded that from a clinical perspective, the selected features appeared plausible and potentially capable of correctly classifying cervical LN in HNSCC patients [5].

Certain limitations of the study need to be addressed. Firstly, the study had a retrospective design and was executed on a single, moderately sized clinical cohort, with only three different labels. From the 1100 potentially eligible patients, only a third met the inclusion criteria. In addition, to reduce the high workload of manually segmenting LNs the sample size was reduced by randomly sampling 28 patients. More data diversity would have allowed stronger conclusions about the general differences in the power of the evaluated feature selectors. Secondly, all patients enrolled in the study were treated with primary concurrent radiochemotherapy. Histopathologic confirmation of the LN labels, which was not available for this study, would have significantly improved the study’s quality.

#### 3.2.2. Cystic Cervical Lymph Nodes of Papillary Thyroid Carcinoma, Tuberculosis, and Human Papillomavirus Positive Oropharyngeal Squamous Cell Carcinoma: Utility of Deep Learning in Their Differentiation on CT

In 2021, in a monocentric, retrospective study performed in the United States and in Japan, Onoue and colleagues explored the utility of DL in the differentiation of cystic cervical LNs. The authors’ rationale was that cystic cervical LNs occur in various pathologies of the head and neck including papillary thyroid carcinoma, tuberculosis, and HPV-positive oropharyngeal HNSCC. Thus, the authors hypothesized that DL might aid as a diagnostic support tool in the radiologic differentiation of cystic cervical LNs [34].

Radiological reports of pretreatment contrast-enhanced neck CTs were explored for cystic cervical LNs related to any of the three diseases. Eligible CTs were included if either of the three diagnoses was histopathologically confirmed by subsequent surgery. LNs were selected by correlating radiological reports with surgical and histopathological reports in the electronic medical records of the included patients. Segmentation was performed manually as two-dimensional, peripheral border delineation at the largest LN-area, performed in axial CT scans [34].

For the deep learning analytical process to construct a diagnostic model differentiating among the three disease, MATLAB ver. 2019a (Mathworks, Natrick, MA, USA) was used on the training dataset. For transfer learning, deep convolutional neural network ResNet-101 was used. Diagnostic accuracy was defined as the primary outcome parameter. Two experienced neuroradiologists, blinded to each other and to the histopathologic findings, reviewed all LNs attributed to test data set. The diagnostic accuracy between the DL algorithm and the neuroradiologist was compared [34].

A total of 42 contrast-enhanced neck CT scans with a total of 173 LN met the inclusion criteria. Of these, 19 patients were diagnosed with HPV-positive oropharyngeal HNSCC, contributing a total of 60 LNs to the analysis. Of these, 45 LNs were attributed to the training-set and 15 to the test-set. The diagnostic accuracy of the training session was 0.88 for DL. For the testing session, a diagnostic accuracy of 0.76 for DL was observed, compared to 0.48 and 0.41 for the two radiologists, respectively (*p* < 0.01) [34].

The authors concluded that the DL analysis using ResNet-101 could differentiate among the three groups of pathologic cervical LN. ResNet-101 exceeded two experienced neuroradiologists. Thus, DL holds promise to become a diagnostic support tool to interpret pathologic cervical LNs [34].

The strengths of the study include its application of a sophisticated deep convolutional neural network and the definition of histopathologic findings of neck dissection as reference for the LN classification. In addition, all suspicious LN were included in the study to minimize selection bias. Limitations include the retrospective, single center study design, and the small number of HNSCC patients included in the study’s limitation to HPV-positive oropharyngeal HNSCC only. The performance level of deep learning analysis could have been higher, and the results more generalizable with a larger sample size and a smaller number of lymph nodes selected per scan. In addition, no separate calculation of diagnostic accuracy parameters for HNSCC LNs only was performed.

#### 3.2.3. Nodal-Based Radiomics Analysis for Identifying Cervical Lymph Node Metastasis at Levels I and II in Patients with Oral Squamous Cell Carcinoma Using Contrast-Enhanced Computed Tomography

In 2021, in a retrospective, monocentric study performed in Japan, Tomita and co-workers investigated whether CT-based texture analysis with machine learning can accurately distinguish pathologic from benign cervical LNs in patients suffering from HNSCC of the oral cavity [35].

Electronic medical records were explored for patients with HNSCC of the oral cavity, who underwent neck dissection. Eligible patients were included if pretreatment contrast-enhanced neck CTs and histopathologic results of neck dissection was available. Correlation of LNs in pretreatment CTs, intraoperatively and in the histopathologic specimen was performed by applying the AHNS cervical LN level system: each cervical LN was set by the surgeons to adjust their relative position with reference to the size and location of the LN in pretreatment CTs. Intraoperatively, the LNs were labeled by the surgeon and thereafter correlated with histopathology. Segmentation was performed manually as three-dimensional, peripheral border delineation across all slices, performed in axial CT scans using LIFEx Software by two experience radiologists [35].

In the texture features with significant differences between benign and pathologic cervical LNs, the best combined texture features were selected using recursive feature elimination, which limited the number of important features to the most important ones. A Python-based support vector machine (SVM) with radial-basis functional kernel in scikit-learn (v.016.1; https://scikit-learn.org, accessed on 1 October 2022.) was implemented in the training session to evaluate the best models for the validation session. Diagnostic accuracy, sensitivity, specificity, and the AUC were defined as primary outcome parameters. Two experienced radiologists and one experienced maxillofacial surgeon, blinded to each other and the histopathologic findings, reviewed all LNs attributed to the test data set based on the established criteria. An a priori LN sample size calculation was performed using a type I error of 5% and power of 80% resulting in a minimum sample size of 27 with a 3:1 ratio of non-pathologic to pathologic cervical LNs [35].

A total of 23 patients suffering from HNSCC of the oral cavity with a total of 201 LN meet the inclusion criteria. Of these, 150 LN were non-pathologic and 51 pathologic, thus meeting the predefined sample size calculations. Of these, 141 LNs were attributed to the training-set and 60 to the test-set. The diagnostic accuracy of the training session ranged from 0.90 to 0.91 for SVM-based ML. For the testing session, the diagnostic accuracy for SVM-based ML ranged from 0.87 to 0.90, compared to 0.70 to 0.83 for the two radiologists and the one maxillofacial facial surgeon, respectively (all *p* < 0.05) [35].

The authors concluded that SVM-based ML using texture-based features was observed to differentiate between non-pathologic and pathologic cervical LNs in patients suffering from HNSS of the oral cavity [35].

The strengths of the study include its a priori sample size calculation and elaborate methods to correlate histopathologic finding of neck dissection with the radiologic findings of pretreatment CTs. The limitations include the retrospective, single center study design and the use of ML instead of more advanced DL techniques. In addition, the ROIs were manually placed on the cervical LNs. Moreover, texture analyses were calculated in some patients who underwent neck dissection after resection of primary OSCC. The influence of postoperative LN changes may not be excluded, but upon evidence of postoperative inflammatory lesions at least 2–3 months after primary surgery, a postoperative evaluation by the cross-sectional imaging is recommended.

#### 3.2.4. Attention Guided Lymph Node Malignancy Prediction in Head and Neck Cancer

In 2021, in a monocentric, retrospective study, performed in the United States of America and China, Chen and colleagues proposed an attention-guided convolutional neural network (agCNN), which had the primary aim of facilitating LN classification in patients suffering from oropharyngeal HNSCC via two mechanisms: (1) alleviating the critical requirement of large training samples by DL approaches and (2) facilitating the LN segmentation process itself by automated segmentation of LNs after defining 2 ROIs based on multiple activation maps [36].

Patients suffering from oropharyngeal HNSCC, who underwent neck dissection and from whom preoperative contrast-enhanced PET-CTs were available, were eligible. After manual contouring of LNs by an experienced radiation oncologists in the contrast-enhanced CT, the proposed agCNN puts multiple activation maps out of a predefined ROI, whose voxel values indicate the importance of the voxel for the final malignancy prediction. Defining a discriminative region within the ROI, the agCNN minimizes the difference between the multiple activation maps and reference segmentation, ultimately performing automated LN segmentation. In addition, a second classification CNN (cCNN) minimizes the difference between the predicted probability for malignancy and the actual histopathologic nodal status. Both CNNs were trained simultaneously. Diagnostic accuracy, sensitivity, specificity, and the AUC, to classify cervical LNs as benign or malignant, were defined as the primary outcome parameters [36].

A total of 129 patients suffering from oropharyngeal HNSCC with a total of 791 LN met the inclusion criteria. Of these, 620 LN were benign and 171 malignant. The LN data were randomly partitioned in 5 folders with 25–26 LNs per folder. Then, a 5-folder cross validation was performed; 3 folders’ data were used to train the models, 1 folder was used to tune hyperparameters to validate the models, and 1 folder was used to test the models. A total of 5 iterations were performed by the authors. Thereby, every folder was used for testing. A diagnostic accuracy of 0.92 was observed [36].

The authors concluded that the proposed agCNN can highlight the discrimination region in an image of LN malignancy, outperforming conventional CNSS that require accurate segmentation in terms of LN classification [36].

Strengths of the study include its application of sophisticated DL methodology, the definition of histopathology as reference, the comparatively larger sample size, and the 5-fold cross-validation.

#### 3.2.5. Predicting Lymph Node Metastasis in Patients with Oropharyngeal Cancer by Using a Convolutional Neural Network with Associated Epistemic and Aleatoric Uncertainty

In 2020, in a single center, retrospective study performed in the United States, Dohopolski and co-authors assigned measures of uncertainty to a CNN aiming at improving the accuracy of LN classification in patients suffering from oropharyngeal HNSCC [37].

Patients suffering from oropharyngeal HNSCC, who underwent neck dissection and from whom preoperative contrast-enhanced PET-CT were available, were eligible. After manual contouring of the LNs by an experienced radiation oncologists on contrast-enhanced CT, LNs were labeled according to the pathology reports. An AlexNet-like CNN was trained to classify LNs as malignant or benign. Via dropout variational inference and test-time augmentation, the levels of uncertainty were estimated. The CNN performance was stratified accordingly using the validation cohort. For the test cohort, sensitivity, specificity, and the AUC to correctly classify LNs as benign or malignant were defined as primary outcome parameters [37].

A total of 129 patients suffering from oropharyngeal HNSCC with a total of 791 LN meet the inclusion criteria. Of these, 620 LN were benign and 171 malignant. The LN data were randomly assigned to the training (*n* = 479), validation (*n* = 125), or test cohort (*n* = 187). Sensitivity and specificity for the validation cohort ranged from 0.67 to 1.0 and 0.37 to 0.98, respectively, with higher sensitivities and specificities observed for lower levels of uncertainty. Although not explicitly reported by the authors, this translates to diagnostic accuracies of 0.43 to 0.99 for the validation cohort. For the testing cohort, the model achieved a sensitivity and specificity of 0.94 and 0.90, respectively. Although not explicitly reported by the authors, this translates to a diagnostic accuracy of 0.91 for the test cohort.

The authors concluded that assigning measures of uncertainty to CNN predictions could improve the accuracy of LN classification in oropharyngeal HNSCC patients by efficiently identifying instances where expert evaluation is needed to corroborate a model’s prediction [37].

The strengths of the study include its application of sophisticated DL methodology and the comparatively large sample size. However, the various limitations of the study ought to be discussed. Firstly, correlating the written pathology report to PET-CT images involved a degree of subjective judgment. The authors only included data with a high degree of certainty. Any uncertain LNs were reviewed by a senior radiation oncologist and were excluded if any doubt remained. It is likely that this process biased the data and led the model to discriminate according to known correlates associated with malignant LNs, such as LNs of large size, LNs with a necrotic appearance, or LNs with high PET avidity. Another limitation includes the lack of calculation of all standardized diagnostic accuracy parameters (e.g., sensitivity, specificity, and AUC).

#### 3.2.6. Multi-INSTITUTIONAL Validation of Deep Learning for Pretreatment Identification of Extranodal Extension in Head and Neck Squamous Cell Carcinoma

In 2019, in a multicenter, retrospective study, performed in the United States of America and Canada, Kann and co-authors validated a previously developed DL algorithm that identifies ECS on pretreatment CT-scans of HNSCC patients from a diverse set of institutions and compared the algorithms’ diagnostic ability to that of expert diagnosticians [38].

Patients suffering from HNSCC of the oropharynx, oral cavity, larynx, hypopharynx, nasopharynx, and salivary glands, who underwent neck dissection and for whom preoperative contrast-enhanced CTs were available, were eligible. The previously developed DL algorithms were externally validated via two data sets: (a) Mount Sinai Hospital (New York, NY, USA) and (b) the Cancer Genome Atlas HNSCC imaging data, which compiles data from seven different institutions. All LNs were manually contoured and labeled as ECS+ or ECS− according to the pathology reports. A DualNet DL-algorithm was previous trained to classify pathologic cervical LNs as ECs+ or ECS− on a data set of 2875 segmented and labeled LNs. For the validation, an a priori sample size of at least 70 LNs was calculated to evaluate the study’s primary end point, which was the AUC of the ROC. In addition, sensitivity, specificity, and diagnostic accuracy for the DL-algorithm to correctly classify pathologic cervical LNs ECS+ or ECS− was provided. The DL-algorithms performance was compared to two experienced head-and-neck radiologists [38].

A total of 144 patients suffering from HNSCC of the oropharynx, oral cavity, larynx, hypopharynx, nasopharynx, and salivary glands with a total 200 LN met the inclusion criteria. Of these, 82 patients with 130 LNs were enrolled from Mount Sinai Hospital and 62 patients with 70 LNs from the 7 different institutions, which were compiled in the Cancer Genome Atlas. Since the training was performed on a separate set of 2875 labeled LNs, all LNs were used as the test cohort (*n* = 200), and therefore, exceeded the a priori sample size calculation of a minimum of 70 LNs. The DL-algorithm achieved an AUC of 0.84 to 0.90, outperforming the radiologists (AUC 0.60–0.70 and 0.71–0.82; all *p* < 0.02, respectively). This translates to a diagnostic accuracy of 83.1–88.6 for the DL-algorithm compared to 73.8–88.6 for the radiologists (all *p* < 0.02) [38].

The authors concluded that their previously developed DL-algorithm is capable of successfully identifying ECS in pretreatment CT scans across multiple institutions and tumor sites, exceeding the diagnostic abilities of experienced head-and-neck radiologists [38].

Although the present study was well designed, several limitations need to be addressed. Firstly, the time from pretreatment CT-scan to neck dissection, from which the histopathologic finding was defined as reference, was not standardized. Secondly, both the training and the validation dataset primarily consisted of comparatively large LNs. Thus, the DL algorithm explored can only be applied with caution to smaller LNs. In addition, the authors mentioned that their study was underpowered to predict a statistically significant improvement compared to the radiologists’ report.

#### 3.2.7. Dual-Energy CT Texture Analysis with Machine Learning for the Evaluation and Characterization of Cervical Lymphadenopathy

In 2019, in a monocentric, retrospective study performed in Canada, Seidler and co-authors determined whether ML-assisted texture analysis of multi-energy virtual monochromatic image datasets from dual-energy CT can be used to differentiate pathologic cervical LN from non-pathologic LN in HNSCC-patients amongst others [39].

Patients suffering from HNSCC of an unspecified tumor site who underwent neck dissection and with available preoperative contrast-enhanced dual-energy CTs, were eligible. All LNs were manually contoured around the largest diameter of the LN in axial planes by three experienced radiologists and labeled pathologic or non-pathologic according to pathology reports of neck dissection. Texture analysis was performed using a filtration-histogram technique with commercially available research software TexRad (Cambridge, UK). ML and construction of prediction models was performed based on a training set and a testing dataset for Random Forest- and Gradient Boosting Machine-algorithms. No a priori sample size calculation was performed. Accuracy, sensitivity, specificity, and AUC to distinguish pathologic from non-pathologic cervical LNs in HNSCC patients were defined as the primary outcome parameter [39].

A total of 20 patients suffering from HNSCC of unspecified tumor site with a total of 176 LNs met the inclusion criteria. Of these, 31 LNs were labeled pathologic and 176 non-pathologic. Due to the imbalance between the number of pathologic and non-pathologic cervical LNs, a maximum of 4 LNs per patient with non-pathologic LNs only for pairwise comparison were randomly selected for the 2 ML-algorithms. Thus, 31 pathologic cervical LNs and 39 non-pathologic cervical LNs were selected, of which 70% (=49 LNs) were attributed to the training set and 30% (=21 LNs) to the test set. For the training set, RF and GMB ML-algorithms achieved a diagnostic accuracy to distinguish pathologic from non-pathologic cervical LNs of 96% and 98%, respectively. For the test set, RF and GMB ML-algorithms achieved a diagnostic accuracy of 85% and 90%, respectively [39].

The authors concluded that ML-assisted Dual-Energy-CT texture analysis using TexRad and a RF or GMB-based ML-algorithms aids in distinguishing pathologic from non-pathologic cervical LNs in HNSCC patients [39].

Various limitations of the study need to be addressed. Firstly, the number of included patients and LNs was comparatively small, which is reflected in the large confidence intervals. Secondly, histopathologic findings of neck dissection as reference were only available for patients suffering from HNSCC. Unlike HNSCC, patients included in the study with lymphoma or inflammatory nodes were not biopsied. Consequently, it is possible that there is some unintended bias from the inclusion of multiple pathologic nodes from the same patient. Furthermore, a potential bias related to the recruitment from a single center specialized in cancer treatment is possible.

#### 3.2.8. CT Evaluation of Extranodal Extension of Cervical Lymph Node Metastases in Patients with Oral Squamous Cell Carcinoma Using Deep Learning Classification

In 2019, in a monocentric, retrospective study performed in Japan, Ariji and co-workers aimed at clarifying CTs diagnostic performance in classifying ECS in pathologic cervical LNs of HNSCC patients of the oral cavity using a DL-algorithm [40].

Patients suffering from HNSCC of the oral cavity, which underwent neck dissection and with available preoperative contrast-enhanced CTs, were eligible. Instead of LN segmentation, a two-dimensional axial CT-section at the center of the LNS and four to six consecutive axial CT-sections above and below of the center of the LN were manually selected by an experienced radiologist. All pathologic cervical LNs were labeled as either with or without ECS according to the pathology reports of neck dissection. A total of 80% of the images were assigned to the training set, for which DIGITS DL-algorithm was used and 20% of the images were assigned to the test set, for which AlexNet DL-algorithm was used. The selection of the images was performed automatically. No a priori sample size calculation was performed. Diagnostic accuracy to correctly classify LNs as “pathologic with ECS” was defined as the primary outcome parameter. The DL-algorithms performance was compared to four experienced radiologists applying established, shape-based ECS criteria (i.e., minor axis, central necrosis, and irregular borders) [40].

A total of 51 patients suffering from HNSCC of the oral cavity with a total of 143 pathologic cervical LNs were included. Of these, 33 LNs were labeled “pathologic with ECS” and 110 “pathologic without ECS”. Of the 143 pathologic cervical LNs, 80% (*n* = 114) were automatically assigned to the training set and 20% (*n* = 29) to the test set. The DL-algorithm’s diagnostic accuracy was 84.0% compared, which was significantly higher than the radiologists’ diagnostic accuracies ranging from 51.1% to 62.6%, applying shape-based ECS criteria [40].

The authors concluded that their DL diagnostic performance in distinguishing ECS was significantly higher than that of radiologists using shape-based ECS criteria [40].

The main limitation of the present study was its single institutional design and the small sample size, which is prone to be underpowered.

#### 3.2.9. Combining Many-Objective Radiomics and 3D Convolutional Neural Network through Evidential Reasoning to Predict Lymph Node Metastasis in Head and Neck Cancer and Predicting Lymph Node Metastasis in Head and Neck Cancer by Combining Many-Objective Radiomics and 3-Dimensioal Convolutional Neural Network through Evidential Reasoning

Originally in 2018 [41], and again in 2020, in a monocentric, retrospective study performed in the United States and China, Chen and co-workers aimed at exploring PET-CTs diagnostic accuracy in predicting LN metastases of HNC patients of the oral cavity and pharynx using a hybrid DL-algorithm combining a 3D-CNN and many-objective-radiomics [42].

Patients suffering from HNC of the oral cavity and pharynx, who had previously enrolled in the INFIELD trial between 2016 and 2018, were included. LNs in pretreatment PET-CTs were manually contoured by a radiation oncologist and a nuclear medicine physician. Thereafter, the manually contoured LNs were labeled as normal, suspicious, or involved. No a priori sample size calculation was performed. No histopathologic report of neck dissection was defined as reference for the LN labels. Diagnostic accuracy to correctly classify LNs was defined as the primary outcome parameter. The diagnostic accuracy of the hybrid DL-algorithm combining 3D-CNN and many-objective-radiomics were compared to 3D-CNN and many-objective-radiomics alone [42].

A total of 59 patients suffering from HNC of the oral cavity and pharynx with a total of 236 LNs were included, of which 107 were labeled as involved. Of these, 170 LNs were assigned to the training set and 66 LNs to the testing set. The diagnostic accuracy of the hybrid DL-algorithm combining 3D-CNN and many-objective-radiomics was 88.0%, which was significantly higher than for 3D-CNN or many-objective-radiomics alone (81.0% and 75.0%, respectively, both *p* < 0.0001) [42].

The authors concluded that their hybrid DL-algorithm provided a more accurate way for predicting LN metastases using PET-CT than 3D-CNN or many-objective-radiomics alone [42]. It is to be noted that the same authors explored the same study aim with the same hybrid DL-algorithm, additionally applying evidential reasoning. The diagnostic accuracies observed did not differ from the findings of their study without applying evidential reasoning [42].

One of the main limitations of this study is the small number of patients included in the study. Moreover, the same set of patients and LNs was explored in two different studies with virtually no differences observed in diagnostic accuracy parameters. Moreover, the authors themselves restrained from highlighting these limitations in either of the two studies themselves.

#### 3.2.10. Contrast-Enhanced Computed Tomography Image Assessment of Cervical Lymph Node Metastasis in Patients with Oral Cancer by Using a Deep Learning System of Artificial Intelligence

In 2018, in a monocentric, retrospective study, Ariji and co-workers aimed at investigating CT’s diagnostic performance in classifying LNs in HNSCC patients of the oral cavity using a DL-algorithm [43].

Patients suffering from HNSCC of the oral cavity, which underwent neck dissection and with available preoperative contrast-enhanced CTs, were eligible. Instead of LN segmentation, a two-dimensional axial CT-section at the center of the LNS and 3 mm above and below the center of the LN in consecutive axial CT-sections was manually selected by an experienced radiologist. All LNs were labeled pathologic or non-pathologic according to the pathology reports of neck dissection. The images were randomly assigned to the training set, validation set, and testing set via 5-fold-cross validation. The authors ensured that the training and testing data did not contain samples from the same LN or same patients and that a balanced number of positive and negative samples in each group was maintained. No a priori sample size calculation was performed. Diagnostic accuracy to correctly classify LNs a pathologic or non-pathologic was defined as the primary outcome parameter. The performance of the DL-algorithms was compared to two experienced radiologists applying established criteria (i.e., minor axis, central necrosis, and irregular borders) [43].

A total of 54 patients suffering from HNSCC of the oral cavity with a total of 441 histopathologically proven pathologic and non-pathologic cervical LNs were included. Of these, 127 LNs were pathologic and 314 non-pathologic. The DL-algorithms diagnostic accuracy was 78.2% compared to 83.1% for the two experienced radiologists (*p* = 0.0507) [43].

The authors concluded that the explored DL-algorithm yielded diagnostic results similar to those of the radiologists, which suggests that this system may be valuable for diagnostic support [43].

This study had some limitations. The explored model did not operate in clinical real time because image segmentation was performed manually.

#### 3.2.11. Pretreatment Identification of Head and Neck Cancer Nodal Metastasis and Extranodal Extension Using Deep Learning Neural Networks

In 2018, in a monocentric, retrospective study performed in the United States, Kann and co-authors explored a DL algorithm’s capacity to correctly classify LNs as non-pathologic, pathologic, or pathologic with ECS in pretreatment CT-scans of HNSCC [44].

Patients suffering from HNSCC of the oropharynx, oral cavity, larynx, hypopharynx, nasopharynx and salivary glands, who underwent neck dissection at their institution and from whom preoperative contrast-enhanced CTs were available, were eligible. All LNs were manually contoured slice-by-slice in axial planes and labeled as non-pathologic, pathologic, or pathologic with ECS according to the pathology reports. BoxNet, SmallNet, and DualNet DL-algorithms were trained, tested, and validated. The study’s primary end point was the AUC of the ROC. In addition, sensitivity, specificity, and diagnostic accuracy for the DL-algorithms to correctly classify cervical LNs were provided [44].

A total of 258 patients suffering from HNSCC of the oropharynx, oral cavity, larynx, hypopharynx, nasopharynx, and salivary glands with a total 653 LN met the inclusion criteria. Of these, 153 LNs were pathologic without ECS and 120 were pathologic with ECS. Training was performed on 417 LNs, validation on 105 LNs, and testing on 131 LNs. The DL-algorithm achieved and AUC of 0.91 and 0.91 to correctly classify pathologic LNs with and without ECS, respectively. This translates to a diagnostic accuracy of 85.5% and 85.7% for pathologic cervical LNs with and without ECS, respectively [44].

The authors conclude that their DL algorithm has the potential for use as a clinical decision-making tool to help guide head and neck cancer patient management [44].

Various limitations of the study need to be addressed. Firstly, the process of individual lymph node CT labeling in correlation with pathology reports is subject to some degree of uncertainty and subjectivity. Secondly, the sample size of the study was limited to the number of HNSCC patients undergoing neck dissection at the authors’ institution, possibly resulting in an inferior predictive performance. Additionally, data from several different CT scanner models was used in the training of the DL.

#### 3.2.12. Automatic Detection and Classification of Nasopharyngeal Carcinoma on PET/CT with Support Vector Machine

In 2012, in a monocentric, retrospective study performed in Hong Kong, Wu and colleagues explored the capability of machine learning via support vector machine to automatically detect and classify malignancy—including suspect cervical LNs–in pretreatment PET-CTs of nasopharyngeal HNSCC patients [45].

Both primary tumors and suspect cervical LNs were manually segmented and labeled by experienced nuclear medicine physicians. Candidate lesions were extracted based on features from PET-CT including position, average intensity, area, eccentricity, symmetry, compactness, intensity difference, and textual moments. Additionally, a priori knowledge of anatomical features was considered. The study’s primary end point was the sensitivity to correctly identify malignant, hypermetabolic lesions [45].

A total of 10 patients suffering from nasopharyngeal HNSCC with a total of 25 PET-CTs were included. The ML-algorithm successfully identified all 53 hypermetabolic lesions labeled as malignant, which translates to a sensitivity of 90%. No additional details on diagnostic accuracy parameters, nor on the specific LNs included in the analysis, were provided [45].

The authors concluded that ML applying support vector machine has the potential to accurately classify suspect hypermetabolic lesions in NPC.

One of the main limitations of the study is that no details on the segmented LN were provided. In addition, the number of patients included in the study was small. Thus, the study is likely to be significantly underpowered.

## 4. Discussion

Locally-advanced HNSCC is mainly defined by the presence of pathologic cervical LNs [1]. Current radiologic criteria to classify LNs as pathologic or pathologic with ECS are mainly shape-based [14,15,16,17,18,19]. However, significantly more quantitative information is contained within imaging data that could be exploited for classification of LNs in patients with locally-advanced HNSCC [5]. This quantitative information contained within images represents a wealth of data for analysis using AI. Currently, various reviews exploring the role of AI in head and neck imaging are available [24,26,27,28]. However, reviews specifically addressing the current role of AI to classify LN in patients with head and neck squamous cell carcinoma are sparse. The present work aimed at systematically reviewing original articles applying RPISMA guidelines [29] and the Study Quality Assessment tool of NIH [30], which specifically explore the current role of AI to classify of LNs in patients with locally-advanced HNSCC.

In the last 10 years, 13 studies were identified, which applied AI to classify cervical LNs [5,34,35]. All of the studies were retrospectively designed and were mainly single-center studies. The majority of the studies explored patients with HNSCC of the oropharynx or oral cavity. Only a limited number of studies addressed laryngeal, nasopharyngeal, or salivary gland HNSCC. The most frequently applied imaging modalities were CTs and PET-CTs. None of the identified studies explored the role of MRI to classify cervical LNs via the mean of AI. Despite significant differences in the number of included patients and included LNs, the diagnostic accuracy to correctly classify LNs as pathologic or pathologic with ECS was above 85% with some studies reaching levels of 99% [5,34,35,36,37,38,39,40,41,42,43,44,45].

## 5. Conclusions

All of the included studies concluded AI to be a potentially promising diagnostic support tool for the classification of pathologic cervical LNs and pathologic cervical LNs with ECS. Since all of these studies were retrospectively designed, adequately powered, prospective, and randomized, control trials are urgently required to further assess the role of AI in LN classification in HNSCC patients.

## Figures and Tables

**Figure 1 cancers-14-05397-f001:**
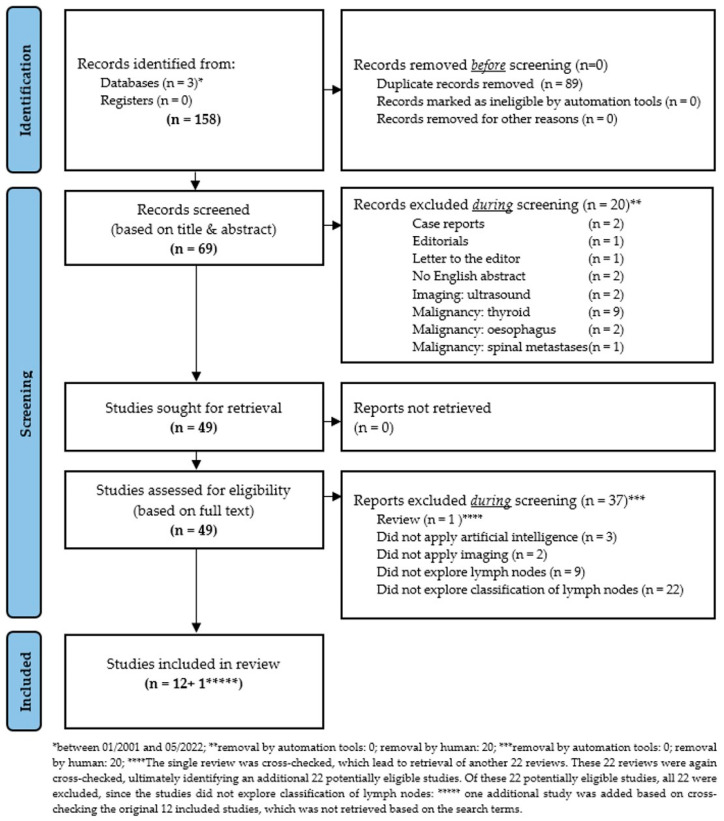
Modified PRISMA 2020 flow diagram for new systematic reviews, which includes searches of databases and registers only, as proposed by Page MJ and co-authors [29].

**Table 1 cancers-14-05397-t001:** Details about the applied criteria to obtain the NIH-score [30].

Criteria	Yes ^1^	No
1. Was the research or objective in this paper clearly stated?	1 point	0 points
2. Was the study population clearly specified and defined?	1 point	0 points
3. Was the participation rate of eligible persons at least 50%?	1 point	0 points
4. Were all the subjects selected or recruited from the same or similar populations? Were inclusion and exclusion criteria for being in the study prespecified and applied uniformly to all participants?	1 point	0 points
5. Was a sample size justification, power description, or variance and effect estimate provided?	1 point	0 points
6. For the analyses in this paper, were the exposure(s) of interest measured prior to the outcome(s) being measured?	1 point	0 points
7. Was the timeframe sufficient, such that one could reasonably expect to see an association between exposure and outcome if it existed? 8. For exposures that can vary in amount or level, did the study examine different levels of the exposure as related to the outcome?	1 point	0 points
9. Were the exposure measures clearly defined, valid, reliable, and implemented consistently across all study participants?	1 point	0 points
10. Was the exposure(s) assessed more than once over time?	1 point	0 points
11. Were the outcome measures clearly defined, valid, reliable, and implemented consistently across all study participants?	1 point	0 points
12. Were the outcome assessors blinded to the exposure status of participants?	1 point	0 points
13. Was the loss to follow-up after baseline 20% or less?	1 point	0 points
14. Were key potential confounding variables measured and adjusted statistically for their impact on the relationship between exposure(s) and outcome(s)?	1 point	0 points

^1^ Points were added and rated as “good” if ≥11 items were rated as “yes”, “fair” if ≥8 items were rated as “yes” or “poor” if ≤7 items rated as “yes”.

**Table 2 cancers-14-05397-t002:** Data extraction sheet.

Author, Year, Country	Tumor Site	Imaging Modality	No. Patients	No. cN+/pN+ LNs	No. LN-Training	No. LNs-Validation	No. LNs-Testing	Type of AI (Subtypes)	Sensitivity	Specificity	Diagnostic Accuracy	AUC	NIH Quality Sum	NIH Quality Grade
Bardosi, 2022, Austria	oral, pharynx, larynx, CUP	CT	28	182 of 252	252	n.a.	n.a.	ML (EFS)	n.a.	n.a.	0.87/0.96	n.a.	8	poor
Onoue, 2021, US, Japan	oropharynx	CT	19	60 of 60	45	n.a.	15	DL (CNN)	n.a.	n.a.	0.76	n.a.	10	fair
Tomita, 2021, Japan	oral	CT	23	51 of 201	141	60	n.a.	ML (SVM)	0.70–0.90	0.95	0.87–0.90	0.82–0.93	12	good
Chen, 2021, US, China	oropharynx	PET-CT	129	171 of 791	3 × 25–26	1 × 25–26	1 × 25–26	DL (agCNN + CNN)	0.91	0.93	0.92	0.98	12	good
Dohopolski, 2020, US	oropharynx	PET-CT	120	171 of 791	479	125	187	DL (CNN)	0.94	0.90	n.a.	0.99	12	good
Kann, 2019, US, Canada	oral, pharynx, larynx, glands	CT	414	38 of 200	2875	270	144	DL (CNN)	0.71–0.82	0.85–0.91	0.83–0.89	0.84–0.90	13	good
Seidler, 2019, Canada	n.a.	DE-CT	20	31 of 176	49	0	21	ML (RF, GBM)	0.89	0.82–0.91	0.85–0.90	0.96–0.97	10	fair
Ariji, 2019, Japan	oral	CT	51	22 of 143	114	0	29	DL (CNN)	0.67	0.90	0.84	0.82	11	good
Chen, 2020, US, China	oral, pharynx	PET-CT	59	107 of 236	170	0	66	DL (CNN + MOR)	n.a.	n.a.	0.88	0.95	11	good
Chen, 2018, US, China	oral	CT	45	127 of 441	4 × 87–88	1 × 87–88	1 × 87–88	DL	0.75	0.81	0.78	0.80	11	good
Ariji, 2018, Japan	oral, pharynx	PET-CT	41	107 of 236	170	0	66	DL	n.a.	n.a.	0.88	0.95	12	good
Kann, 2018, US	oral, pharynx, larynx, glands	CT	270	273 of 653	417	105	131	DL	0.84/0.88	0.87/0.85	0.86/0.86	0.91/0.91	13	good
Wu, 2012, Hong Kong	nasopharynx	PET-CT	10	n.a.	n.a.	n.a.	n.a.	ML (SVM)	0.90	n.a.	n.a.	n.a.	8	poor

agCNN: attention-guided convolutional neural network; AI: artificial intelligence; AUC: area under the curve; cN+: radiologically labeled pathologic cervical lymph node; CNN: convolutional neural network; CT: computed tomography scan; CUP: carcinoma of unknown primary; DE-CT: dual-energy computed tomography scan; DL: deep learning; EFS: eliminative feature selection; ER: evidential reasoning; GBM: gradient boosting machine; MOR: many-objective radiomics; ML: machine learning; n.a.: not applicable; NIH: Study Quality Assessment Tool of National Institute of Health; No.: number of; PET-CT: positron emission tomography with computed tomography scan; pN+: histopathologically proven pathologic cervical lymph node; RF: random forest; SVM: support vector machine; Abbreviations are provided in alphabetic order; NIH classification of study quality: ≥11 positive rating = “good quality, 8–11 positive ratings = “fair quality, and ≤7 positive ratings = “poor quality”.

**Table 3 cancers-14-05397-t003:** Classification of cervical lymph nodes applying artificial intelligence.

Author, Year, Country	Study Type, No. of Centers	Tumor Site	No. of Patients	No. of +LNs (Training, Validation, Test)	Imaging	Segmentation	LN Classifier	Reference	Training Set AI (Accuracy)	Test Set AI (Accuracy)	Conclusion	NIH Quality (NIH Score)
Bardosi, 2022, Austria	retrospective monocentric	pharynx, larynx, CUP	28	182 cN+ of 252 LNs	CT	manual, 3D	HNSCC; cN+, cN-, ECS+	radiologist label	ML EFS (0.96)	ML EFS (0.96)	EFS-algorithms appeared to be useful to retain high diagnostic accuracy with only 10% of the extracted features	Poor (8/14)
Onoue, 2021, US, Japan	retrospective, monocentric	oropharynx	19	60 pN+ of 60 cN+ (45, n.a., 15)	CT	manual, 2D	HNSCC; pN+ vs. thyroid cancer vs. tuberculosis	histopathology of neck dissection	DL (0.88)	DL (0.76)	DL potent support tool to differentiate pathologic cervical LNs	fair (10/14)
Tomita, 2021, Japan	retrospective, monocentric	oral cavity	23	51 pN+ of 201 cN+ (141, 60, n.a.)	CT	manual, 3D	HNSCC; pN+ vs. pN−	histopathology of neck dissection	ML SVM (0.90–0.91)	ML SVM (0.87–0.90)	ML can differentiate pathologic from non-pathologic cervical LNs	Good (12/14)
Chen, 2021, US, China	retrospective, monocentric	oropharynx	129	171 pN+ of 791 cN+ (25 × 3, 25 × 1, 25 × 1)	PET-CT	manual and automated, 3D	HNSCC; pN+ vs. pN−	histopathology of neck dissection	DL agCNN, cCNN, (0.92)	DL agCNN, cCNN (0.92)	agCNN without accurate LN segmentation outperforms conventional CNNs in LN classification	Good (12/14)
Dohopolski, 2020, US	retrospective, monocentric	oropharynx	129	171 pN+ of 791 cN+ (479, 125, 187)	PET-CT	manual, n.a.	HNSCC; pN+ vs. pN-−	histopathology of neck dissection	DL (0.43–0.99)	DL (0.91)	assigning measures of uncertainty to CNN improves accuracy of LN classification	good (12/14)
Kann, 2019, US, Canada	retrospective, multicentric (*n* = 8)	oral cavity, oropharynx, larynx, glands	144	38 ECS+ of 200 pN+ (0, 0, 200)	CT	manual, 3D	HNSCC; ECS+ vs. ECS−	histopathology of neck dissection	DL (0.86)	DL (0.83–0.89)	DL successfully identified ECS on pretreatment CTs	good (13/14)
Seidler, 2019, Canada	retrospective, monocentric	n.a.	20	31 pN+ of 176 cN+ (49, 0, 21)	CT	manual, 2D	HNSCC; pN+ vs. pN−; HNSCC vs. lymphoma vs. inflammatory	histopathology of neck dissection	ML (6 features) (RF 0.96, GBM 0.98)	ML (6 features) (RF 0.85, GBM 0.90)	ML assisted texture analysis aids in distinguishing different nodal pathologies	Fair (10/14)
Ariji, 2019 Japan	retrospective, monocentric	oral cavity	51	33 ECS+ of 143 pN+ (114, 0, 29)	CT	manual, 2D	HNSCC; ECS+ vs. ECS−	histopathology of neck dissection	DL (n.a.)	DL (0.84)	DL diagnostic performance in distinguishing ECS outperforms shape-based ECS criteria	Good (11/14)
Chen, 2020, US, China	retrospective, monocentric	oral cavity, pharynx	59	107 cN+ of 266 LNs (170, 66, 0)	PET-CT	manual, n.a.	HNSCC; cN-, cN+/−, cN+	radiologists label	DL hybrid (0.88)	DL hybrid (0.88)	Hybrid method of DL and many-objective-radiomics provides more accuracy for predicting LN metastases	good (11/14)
Chen, 2018, US, China	retrospective, monocentric	oral cavity, pharynx	59	107 cN+ of 266 LNs (170, 66, 0)	PET-CT	manual, n.a.	HNSCC; cN-, cN+/−, cN+	radiologists label	DL hybrid + ER (0.88)	DL hybrid + ER (0.88)	Hybrid method of DL and many-objective-radiomics with evidential reasoning provides more accuracy for predicting LN metastases	good (11/14)
Ariji, 2018, Japan	retrospective, monocentric	oral cavity	45	127 pN+ of 441 cN+ (1 × 88, 1 × 88, 3 × 88)	CT	manual, 2D	HNSCC; pN+ vs. pN−;	histopathology of neck dissection	DL (0.78)	DL (0.78)	DL yielded diagnostic results similar to those of radiologists	good (12/14)
Kann, 2018, US	retrospective, monocentric	oral cavity, pharynx, larynx, glands	258	273 pN+ of 653 LNs (417, 105, 131)	CT	manual, 3D	HNSCC; pN−, pN+, ECS+	histopathology of neck dissection	DL (0.86)	DL (0.86)	DL has the potential for use as a clinical decision-making tool	Good (13/14)
Wu, 2012, Hong Kong	retrospective monocentric	nasopharynx	10	n.a.; 25 sets of image slices (4:1 ratio training to testing)	PET-CT	manual, 2D	HNSCC; malignant vs. non-malignant	radiologist label	ML SVM (n.a.)	ML SVM (n.a.)	ML has the potential to accurately classify suspect hypermetabolic lesions in NPC	Poor (8/14)

2D: two-dimensional; 3D: three-dimensional; agCNN: attention-guided convolutional neural network; AI: artificial intelligence; AUC: area under the curve; cN-: radiologically labeled non-pathologic cervical lymph node; cN+: radiologically labeled pathologic cervical lymph node; CNN: convolutional neural network; CT: computed tomography scan; CUP: carcinoma of unknown primary; DL: deep learning; ECS−: pathologic cervical lymph node without extracapsular spread; ECS+: pathologic cervical lymph node with extracapsular spread; EFS: eliminative feature selection; ER: evidential reasoning; GBM: gradient boosting machine; HNSCC: head and neck squamous cell carcinoma; LN: cervical lymph nodes; ML: machine learning; n.a.: not applicable; NIH: Study Quality Assessment Tool of National Institute of Health; No.: number of; PET-CT: positron emission tomography with computed tomography scan; pN−: histopathologically proven non-pathologic cervical lymph node; pN+: histopathologically proven pathologic cervical lymph node; RF: random forest; SVM: support vector machine; Abbreviations are provided in alphabetic order; NIH classification of study quality: ≥11 positive rating = “good quality, 8–11 positive ratings = “fair quality, and ≤7 positive ratings = “poor quality”.

## Data Availability

The data presented in this study are available on request from the corresponding author.

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
