# Peer review of "Current Applications of Artificial Intelligence to Classify Cervical Lymph Nodes in Patients with Head and Neck Squamous Cell Carcinoma—A Systematic Review"

_cancers, 2022, doi:10.3390/cancers14215397_

Round 1
Reviewer 1 Report
This is a comprehensive systematic review of studies where AI has been used to assess cervical lymph node involvement in HNSCC. There is good use of figures and tables to cover the identification of eligible studies and summation of the data, although it would have been helpful to have a more detailed coverage of how the NIH Score was obtained.
It is fairly well-written, although there are some grammatical and sentence construction errors evident in places.
Given the growing interest in using AI techniques in tumour assessment then I think this is a timely review which will be of interest to readers and therefore I would be happy to recommend publication if the comments above are addressed.
Author Response
Dear editors, dear referees,
the authors very much appreciate the constructive and positive comments of the referees about the manuscript “Current applications of artificial intelligence to classify cervical lymph nodes in patients with head and neck squamous cell carcinoma – a systematic review” (ID cancers-1985196). All suggested corrections and changes are marked with track changes in the main manuscript.
The authors would like to thank the referees for their time and effort since their suggested changes substantially improved the quality of the manuscript. Thank you very much for reconsidering our manuscript for publication in Cancers.
Referee 1
- “This is a comphrensive systematic review of studies were AI has been used to assess cervical lymph node involvment in HNSCC. There is good use of figures and tables to cover the identification of eligible studies and summation of data.”
All authors involved in this study and the preparation of the manuscript very much appreciate this positive comment.
2. “It would be helpful to have a more detailed coverage of how the NIH Score was obtained.
The authors very much appreciate this constructive suggestion. A more detailed coverage of the process obtaining the NIH was score was included in the manuscript as recommend. Please refer to page 5 of the Methods section, line 152 to 153 and the newly created table 1.
3. “It is fairly well-written, although there are some grammatical and sentence construction errors evident in places”
All authors involved in the preparation of the manuscript very much appreciate this constructive comment. A professional English native-speaker was involved to identify and correct all grammatical errors and sentence construction errors present in the manuscript. The authors are confident that by doing so the readability of the manuscript was significantly improved.

Reviewer 2 Report
The manuscript is well written and comprehensive. However, some minor changes should be addressed:
Row 41. please amend the phrase; you stated “Deep learning was applied in 9 studies and machine learning in 4 of 13 studies”. Please take into consideration that all deep learning is machine learning, but not all machine learning is deep learning; you found 13 studies applying machine learning, 9 of them applying deep learning. These differences are very well explained in your manuscript in paragraph from rows 81-87.
Rows 54-55. Please rephrase “Locally-advanced head and neck squamous cell carcinoma (HNSCC) is mainly defined by the presence of pathologic cervical lymph nodes (LNs)”. Not all locally advanced HNSCC present lymph nodes metastases. “Mainly” is easily miss.
Please follow the rules for listing references as recommended (https://www.mdpi.com/journal/cancers/instructions#preparation):
- Reference numbers should be placed in square brackets [ ] within the text
- Use the recommended format: Author 1, A.B.; Author 2, C.D. Title of the article. Abbreviated Journal Name Year, Volume, page range.
- digital object identifier (DOI) must also be cited when available (https://www.mdpi.com/authors/references)
- For documents co-authored by a large number of persons (more than 10 authors), you can either cite all authors, or cite the first ten authors, then add a semicolon and add ‘et al.’ at the end (https://mdpi-res.com/data/mdpi_references_guide_v5.pdf). You have been citing the first 3 authors.
Author Response
Dear editors, dear referees,
the authors very much appreciate the constructive and positive comments of the referees about the manuscript “Current applications of artificial intelligence to classify cervical lymph nodes in patients with head and neck squamous cell carcinoma – a systematic review” (ID cancers-1985196). All suggested corrections and changes are marked with track changes in the main manuscript.
The authors would like to thank the referees for their time and effort since their suggested changes substantially improved the quality of the manuscript. Thank you very much for reconsidering our manuscript for publication in Cancers.
Referee 2
1. “The manuscript is well written and comprehensiv”
All authors involved in this study and the preparation of the manuscript very much appreciate this positive comment.
2. “Row 41. Please amend the phrase; you stated “Deep learning was applied in 9 studies and machine learning in 4 of 13 studies.”
The authors apologize for this mistake. The sentence in page 1 of the Abstract section, row 41 was changed as suggested.
3. “Rows 54-55. Please rephrase “Locally-advanced head and neck squamous cell carcinoma (HNSCC) is mainly defined by the presence of pathologic cervical lymph nodes (LNs”. Not all locally advanced HNSCC present with lymph node metastases. “Mainly is easily missed.”
All authors involved in the preparation of the manuscript very much appreciate this positive comment. This imprecision was rephrased as suggested. Please refer to page 2 of the Introduction section, row 55 to 56.
4. “Reference numbers should be placed in square brackets [] within the text.
All authors apologize for this formatting mistake. All references numbers, which were originally presented as superscript numbers are now placed in square brackets, as recommend.
5. “Use the recommended format: Author 1, A.B.; Author 2, C.D. Title of article. Abbreviated Journal Name Year, Volume, page range”
All authors apologize for the formatting mistake of the references section. The references section was formatted, as recommend. Please refer to page 20, Reference section, row 680 to 737 and page 21, row 738 to 772.
6. “Digital object identifier (DOI) must also be cited when available”
All authors apologize for not including the DOIs in the references section. DOIs were included in the references section, as recommend. Please refer to page 20, Reference section, row 680 to 737 and page 21, row 738 to 772.
7. “For documents co-authored by a larg number of persons (more than 10 authors), you can either cite all authors, or cite the first ten authors, then add a semicolon and add “et al.” at the end”
All authors apologize for not the correct number of authors for each cited reference in the references section. If a large number of persons co-authored a study, we included all authors’ names, as recommended. Please refer to page 20, Reference section, row 680 to 737 and page 21, row 738 to 772.
